# Association between Prevention from Going Out and Incidence of Falls among Community-Dwelling Older Adults during COVID-19 Pandemic

**DOI:** 10.3390/ijerph20032650

**Published:** 2023-02-01

**Authors:** Saori Anezaki, Mariko Sakka, Noriko Yamamoto-Mitani

**Affiliations:** 1Department of Gerontological Home Care and Long-Term Care Nursing, Division of Health Sciences and Nursing, Graduate School of Medicine, The University of Tokyo, Bunkyo City, Tokyo 113-0033, Japan; 2The Faculty of Medicine, The University of Tsukuba, 1-1-1 Tennodai, Tsukuba City 305-8575, Japan

**Keywords:** COVID-19, community-dwelling older adults, incidence of falls, frequency of going out

## Abstract

To prevent falls, community-dwelling older adults must maintain regular physical activities. This study aimed to explore the association between the prevention from going out and the incidence of falls among community-dwelling older adults during the COVID-19 pandemic. We conducted a prospective cohort study that consisted of 381 individuals aged 65 years or older, living in a local community in Japan, and ranging from being independent to being physically and cognitively frail. The finding revealed that among those who had been going out five or more times weekly pre-pandemic, the prevention from going out at the time of the first state of emergency (SOE) (AOR = 6.84; 95%CI = 1.51–31.02), having history of falls (AOR = 7.35; 95%CI = 1.81–29.84), participating in group gatherings (AOR = 6.09; 95%CI = 1.48–25.12), living with spouses (AOR = 0.08; 95%CI = 0.02–0.40), and living with other than spouse (AOR = 0.15; 95%CI = 0.03–0.73) were associated with the incidence of falls. The study highlights the importance of providing regular opportunities to go out to community-dwelling older adults in order to prevent falls.

## 1. Introduction

In recent years, Japan has become a hyper-aged society with a rapid increase in the older adult population. The percentage of the population aged 65 years and above was 29.1% in 2021 [1], which might rise further to 35.3% by 2040 [2]. There is an increasing concern regarding the rise in social welfare benefit costs related to medical and long-term care [3]. Therefore, the Japanese government is taking an initiative to promote the extension of healthy life expectancy to achieve a society where people can live wholesome and enriched lives with limited financial resources to reduce medical expenditure [4].

To increase healthy life expectancy among community-dwelling older adults, they must maintain their physical and cognitive function. According to a comprehensive survey of living conditions conducted by the Japanese government, falls are the leading cause of the need for long-term care among older adults, excluding unavoidable factors such as dementia [5]. A fall is “a person unintentionally coming to the ground or some lower level and other than as a consequence of sustaining a violent blow, loss of consciousness, sudden onset of paralysis as in stroke or an epileptic seizure” [6]. There is a 10–20% [7,8,9] annual incidence of falls among community-dwelling older adults in Japan. Approximately 20% of the total falls result in injuries requiring medical treatment, and around 5–10% involve fractures [10]. Fractures, in general, are associated with a higher rate of subsequent impairment in activities of daily living and a higher rate of institutionalization [11,12]. Furthermore, regardless of injury type, falls are associated with a loss of confidence in walking and the fear of falling again [13]. When community-dwelling older adults experience fear of falling, they become reluctant to move and decrease their physical activity [14]. Accordingly, their physical function declines, and they are more likely to fall, which can result in the need for long-term care. Therefore, irrespective of the consequences, avoiding falls is essential in older adults regardless of the consequences.

To prevent falls, community-dwelling older adults must maintain regular physical activity. Systematic reviews have described the importance of maintaining physical activity as a risk factor for falls in community-dwelling older adults [15]. Decreased muscle strength, balance, and ability to perform activities of standing and walking resulting from decreased physical activity are associated with the incidence of falls [16,17]. Moreover, physical activities in community-dwelling older adults can be maintained or improved through training and lifestyle modifications [18]. To maintain their physical activity, it is necessary to focus on low-intensity activities such as shopping [19,20]. Previously, opportunities for community-dwelling older adults to go out, such as exercise classes, have been encouraged, but the COVID-19 pandemic has considerably changed the situation.

On 28 January 2020, the COVID-19 outbreak in Japan was first confirmed in a person with no history of overseas travel. Subsequently, the first state of emergency (SOE) was issued on 7 April 2020, for seven prefectures, including Tokyo, Kanagawa, and Saitama. On 16 April, the target area was expanded to all prefectures. Infections spread rapidly in the months following the confirmation of the initial infection, as it occurred worldwide. It is worth noting that the declaration of the SOE is not an enforceable “lockdown”, as in other countries, but rather a humble request to residents to refrain from unnecessary going out while maintaining necessary economic and social services, such as public transportation. Although not mandatory, many Japanese community-dwelling older adults in Japan have thus refrained from going out.

Consequently, community-dwelling older adults in Japan have reported decreased muscle strength because of being unable to go out during the COVID-19 pandemic [21]. Comparing the pre-pandemic period to the time of the first SOE, the number of respondents who reported anxiety about falling or being unable to walk in the future has increased significantly [22]. Therefore, community-dwelling older adults may have an increased risk of falling due to interrupted going out during the COVID-19 pandemic, resulting in decreased physical activity. Thus, this study aimed to examine the association between the prevention from going out during the COVID-19 pandemic and the incidence of falls among community-dwelling older adults.

## 2. Materials and Methods

### 2.1. Study Design

This is a prospective cohort study.

### 2.2. Setting

We conducted this research in Nagatoro town, in a mountainous region of Saitama prefecture near Tokyo. The population was approximately 7300, of which 35.7% were 65 years or older, and 18.0% were 75 or older, which was higher than the national percentage [23]. Nagatoro town has a high rate of participation in group gathering, which provides an opportunity for community-dwelling older adults to go out. A group gathering is a place or opportunity for various activities at least once a month to prevent frailty [24]. In 2018, the national average participation rate in a group gathering at least once a week was 2.2%, while in Nagatoro town, it was 19.6% [25]. The main activities of group gatherings in Nagatoro town are exercise classes to prevent long-term care, which are held at 13 locations to enable community-dwelling older adults to walk there.

### 2.3. Participants and Survey Procedures

To collect information on community-dwelling older adults pre-pandemic, we used the data from a survey conducted in Nagatoro town in January 2020 [26]. Participants were individuals residing in Nagatoro town who were: (a) aged 65 years or older and (b) independent (i.e., “no care/support needed”) to having physically or cognitively frail in 2020 (i.e., categorized as “support needed” by the long-term care insurance system) [27]. Town officials randomly sampled 900 individuals from 2600 residents aged 65 or older through the long-term care insurance records managed by the town in 2020.

To collect data at the time of the first SOE, we conducted a self-administered questionnaire survey in August 2021, using questionnaire created for this study. We developed the questionnaire by adapting measures from the survey conducted in 2020 and adding a few measures. For this survey, we included the same participants as the 2020 survey. Before mailing the questionnaires, we excluded some participants who had died or entered a nursing facility between January 2020 to August 2021. Furthermore, we asked the participants to remember the time of the first SOE and complete the questionnaire by themselves or with the help of a family member. The participants returned their questionnaires to the researchers via mail. To anonymize the data, IDs were assigned to participants by town officials, which were then linked to their corresponding data.

### 2.4. Measures

#### 2.4.1. Incidence of Falls

In the questionnaire, we asked, “Have you experienced a fall in the past year?” Per the 2021 survey, a fall is when you unexpectedly trip or stumble and end up on your elbows, knees, or hands. The response options were 1 = fell many times, 2 = fell once, to 3 = no fall. In this study, we classified one or more falls as having a fall. This is because the experience of one fall can lead to the fear of falling again [13], which can suppress subsequent physical activity, such as going out, thus further leading to falls. For the analysis, we excluded participants who did not respond to this question.

#### 2.4.2. Prevention from Going out at the Time of the First SOE

We used the 2020 and 2021 surveys to describe the variable “prevention from going out” for this study. To assess the frequency of going out pre-pandemic and at the time of the first SOE, we asked, “Do you go out at least once a week?” The response options were: 1: less than once, 2: once a week, 3: go out 2–4 times a week, to 4: 5 or more times a week. Respondents who reported going out at least once a week pre-pandemic but less than once at the time of the first SOE were categorized as experiencing prevention from going out. In this study, to examine the hypothesis that community-dwelling older adults who experienced prevention from going out at the time of the first SOE are at a higher risk of falling, we excluded those who answered that they went out less than once pre-pandemic from the analysis. Participants who did not respond to this question at one or both time points were also excluded.

#### 2.4.3. Demographic Characteristics in the 2020 Survey

Participants’ demographic characteristics included age, sex, living arrangements, need of daily life support, financial status, history of falls, fear of falls, frequency of going out compared to last year, subjective sense of health, physical function, cognitive and oral function impairment, instrumental activities of daily living (IADL), psychological condition, and participation in community activities.

Financial status was assessed using a question on a five-point Likert-type scale ranging from 1 (very unsatisfied) to 5 (very satisfied). We grouped statements 1–2 and 3–5.

Regarding having history of falls, we asked about the history of falls in the past year. We classified one or more falls as having a fall.

Fear of falling was assessed using a question on a five-point Likert-type scale ranging from 1 = very fearful, 2 = somewhat fearful, 3 = not very fearful, to 4 = not fearful at all. We grouped statements 1–2 and 3–4.

Frequency of going out compared to last year was assessed using a question on four-point Likert-type scale ranging from 1 = very much less, 2 = less, 3 = not much less, to 4 = not less. We grouped statements 1–2 and 3–4.

Subjective sense of health was assessed using a question on a four-point Likert-type scale ranging from 1 = very good, 2 = good, 3 = not so good, to 4 = not good. We grouped statements 1–2 and 3–4.

Physical function was assessed using two items: climbing the stairs without touching the handrail or walls and standing up from a chair without holding onto anything. The response options were: 1 = I can, 2 = I can but have not, to 3 = I cannot. We grouped statement 1 and 2–3.

Cognitive and oral function impairment were assessed using the questions, “Do you often feel forgetful?” and “Do you often experience difficulty chewing?”. The response options were yes or no.

IADL was assessed using two items: going out by themselves using public transportation and shopping for groceries. The response options were: 1 = I can, 2 = I can but have not, to 3 = I cannot. We grouped statements 1 and 2–3.

Psychological condition was assessed using a question: “Have you been feeling disinterested or unable to truly enjoy things?” The response options were yes or no.

Participation in community activities was assessed using four items: volunteers’ club (e.g., welfare activities such as cleaning of public places), group exercises (e.g., ground golf), recreation groups (e.g., crafts and arts), and group gathering (e.g., exercises to prevent frailty). For each item, we asked, “How many times do you participate in this community activity?” We accepted multiple answers to these questions, and the choices given were: 1 = four or more times a week, 2 = two to three times a week, 3 = once a week, 4 = one to three times a month, 5 = few times a year, to 6 = have not participated. We grouped statements 1–4 and 5–6.

#### 2.4.4. Demographic Characteristics in the 2021 Survey

The following three items, identified from previous studies as related factors to the incidence of falls but not included in the 2020 survey, were asked in the 2021 survey.

Referring to the stairs or hills in front of their home, we asked “Do you need to take stairs or steep hills when leaving your home?” The response options were yes or no.

Regarding medications, we asked, “Have you been taking any medication daily for more than 1.5 years?” We accepted multiple answers to this question, and the choices for response were: 1 = none, 2 = drugs to lower high blood pressure (antihypertensive drugs), 3 = drugs to help the patient sleep (sleeping pills), 4 = drugs to help the patient urinate (diuretics), to 5 = drugs to make the patient feel better (antidepressants and anti-anxiety drugs).

Regarding subjective symptoms, we asked, “Have you felt any symptoms for more than 1.5 years that would cause inconvenience to you in daily life?” We accepted multiple answers to this question, and the choices given were: 1 = none, 2 = difficulty seeing, 3 = wobbling, 4 = knee/back pain, 5 = difficulty hearing, 6 = headache, to 7 = other.

### 2.5. Pilot Test

Using the kinship method, we recreated 16 community-dwelling older adults aged 65 years or older who were independent in their daily lives. We conducted a pilot test using the questionnaire checklist [28], which asked about the response time, visibility of text, clarity of question meaning, choices, and quantity of questions. Based on these responses, the questionnaire was modified to ensure its face validity.

### 2.6. Data Analysis

#### Selection of Participants for Analysis

Of the participants who responded to both the 2020 and 2021 surveys, those who could be linked by ID were able to cooperate in answering the questionnaire, responded to the item on the frequency of going out at both time points, and responded to the item on falls in the 2021 survey, were included in the current analysis.

We first performed descriptive analyses of the participants’ characteristics for this study. Bivariate analyses between the incidence of falls and other variables were then performed using Welch’s *t*-test, χ^2^ test, and Fisher’s exact test. Third, a multiple logistic regression analysis using the forced-entry method was conducted to investigate which factors were associated with the incidence of falls. The independent variables were those significantly associated with the incidence of falls in the bivariate analyses (Table 1) and prevention from going out. When entering the variables, Spearman’s rank correlation coefficient was used to ensure that there was no strong correlation between the variables. Finally, to examine the association between a more substantial decrease in the frequency of going out and the incidence of falls, the analysis was conducted only among participants who were weekly going out five or more times before the COVID-19 pandemic. All analyses were performed using IBM SPSS Statistics 28 for Macintosh.

### 2.7. Ethical Considerations

An opt-out document was posted in the town hall for three months regarding the use of data from the survey conducted by the Nagatoro Town officials in 2020. It was also posted on the university website to which the researcher belongs. Along with the questionnaire, each participant was provided with a letter explaining the study’s purpose, methods, privacy protection, and voluntary participation status. We included a question about research participation in the questionnaire and obtained their consent to participate in the study. This study was approved by the Human Research Ethics Committee of the University of Tokyo, Japan (2021133NI-2). This committee grants studies in accordance with the provisions of the Declaration of Helsinki.

## 3. Results

### 3.1. Characteristics of the Participants

A total of 418 respondents whose IDs could be linked responded to both the 2020 and 2021 surveys. We excluded nine persons whose responses had missing items related to the frequency of going out in either the 2020 or 2021 survey and 15 who had missing items pertaining to falls in the 2021 survey. Moreover, we also excluded 13 respondents who indicated that they went out less than once in the 2020 survey, leaving 381 valid respondents. Table 1 summarizes the participants’ characteristics. The mean age and standard deviation were 74.4 ± 6.2 years, and 51.4% were female.

### 3.2. The Frequency of Going out and Incidence of Falls Pre-Pandemic and at the Time of the First SOE

Table 2 shows the frequency of weekly going out. At the time of the 2020 survey, before the COVID-19 pandemic, 154 participants (40.4%) went out five or more times. Of those, 25 (16.2%) went out less than once, 38 (24.7%) went out once, 64 (41.6%) went out 2–4 times, and 27 (17.5%) went out five or more times at the time of the first SOE.

Table 3 summarizes the incidence of falls. Among the 381 participants, 91 (23.9%) had fallen. Of these, 25 (27.5%) went out less than once, 29 (31.9%) went out once, 32 (35.2%) went out 2–4 times, 5 (5.5%) went out five or more times at the time of the first SOE.

Among the 154 participants who had gone out five or more times weekly pre-pandemic, 35 had fallen. Of these, nine (25.7%) went out less than once, nine (25.7%) once, 12 (34.3%) 2–4 times, and five (14.3%) five or more times.

### 3.3. Factors Related to the Incidence of Falls

The association between each independent variable and the incidence of falls was examined using bivariate analyses, presented in Table 1. The incidence of falls was associated with older age (*p* = 0.005), living alone or with other than spouse (*p* < 0.001), need of daily life support (*p* = 0.006), history of falls (*p* < 0.001), worse subjective health (*p* < 0.001), inability to climb stairs (*p* < 0.001), inability to go out by themselves (*p* = 0.008), disinterested in the past month (*p* = 0.044), participation in a group gathering (*p* = 0.012), taking diuretics (*p* = 0.014), and having knee or back pain (*p* = 0.007).

The results of the multivariate logistic regression analysis are presented in Table 4. The incidence of falls was positively but not significantly associated with the prevention from going out at the time of the first SOE (adjusted odds ratio (AOR) = 1.95; 95% confidence interval (CI) = 0.76–5.04. A significant positive association was found between history of falls (AOR = 14.82; 95%CI = 5.78–38.00). However, there was a negative association between living with spouses (AOR = 0.17; 95%CI = 0.05–0.61) and living with other than spouse (AOR = 0.24; 95%CI = 0.07–0.86) compared to living alone.

For the current study, we focused on 154 participants who had gone out five or more times weekly pre-pandemic. The results of the bivariate analysis revealed that falls were associated with older age (*p* = 0.002), living alone or with other than spouse (*p* = 0.002), history of falls in the past year (*p* < 0.001), fear of falling (*p* = 0.025), inability to climb stairs (*p* = 0.006), and participation in group gatherings (*p* = 0.014) (Appendix A). The results of the multivariate logistic regression analyses are presented in Table 5. The incidence of falls was positively associated with prevention from going out at the time of the first SEO (AOR = 6.84; 95%CI = 1.51–31.02), having history of falls (AOR = 7.35; 95%CI = 1.81–29.84), and participation in group gatherings (AOR = 6.09; 95%CI = 1.48–25.12). However, a negative association was found between living with spouses (AOR = 0.08; 95%CI = 0.02–0.40) and living with other than spouse (AOR = 0.15; 95%CI = 0.03–0.73).

## 4. Discussion

The current study examined the association between the prevention from going out at the time of the first SOE and the incidence of falls among community-dwelling older adults. The results revealed a trend toward a positive association between the prevention from going out and the incidence of falls at the time of the first SOE; however, the difference was not statistically significant. For participants who had been going out five or more times per week pre-pandemic, the prevention from going out at the time of the first SOE was positively associated with the incidence of falls.

For participants who had been going out at least once per week pre-pandemic, a trend toward a positive association was observed between the prevention from going out at the time of the first SOE and the incidence of falls; however, the difference was not statistically significant. The group that faced a prevention from going out included those who had a slight decrease in the frequency of going out, from an initial low frequency of a few times a week pre-pandemic to less than once at the time of the first SOE. Therefore, the slight decrease in the frequency of going out may not have been sufficient to be associated with the incidence of falls.

Among the participants who had been going out five or more times per week pre- pandemic, prevention from going out at the time of the first SOE was associated with the incidence of falls. A previous study reported that during the pandemic, people decreased their frequency of going out and spent more time in a sedentary position [29]. A previous study comparing the muscle mass of community-dwelling older adults pre-pandemic, and after the first wave in April–May 2020, revealed a decrease in trunk muscle mass [21]. This study suggested that the prevention from going out at the time of the first SOE may have caused muscle weakness among those who had been going out five or more times per week pre-pandemic, which was significantly associated with the incidence of falls.

Moreover, the participants in this study who went out five or more times per week pre-pandemic may have had a higher subjective sense of health. The frequency of going out is also related to psychosocial aspects, with a higher frequency associated with a higher subjective sense of health [30]. It is thus possible that some of them fell because they stumbled over the slightest step or failed to avoid obstacles, as they perceived themselves as healthy and were unaware of the significant muscle weakness associated with their prevention from their going out.

Among the participants who had been going out five or more times weekly pre-pandemic, participation in a group gathering for frailty prevention during the pre-pandemic was positively associated with falls. A previous study reported that participants’ muscle mass increased during their exercise program. However, after the program, the participants sometimes did not continue their exercise, which led to the disappearance of the program’s effects [31]. Nevertheless, the participants in this study might have lost the opportunity to exercise when the group gathering stopped due to the pandemic. Lack of opportunities for exercise may have decreased muscle strength and balance, leading to falls. Furthermore, a previous study reported that participants in a group gathering had a higher risk of needing long-term care [32]. In the study area, public health nurses and resident volunteers directly approached those who needed care prevention support and encouraged them to participate in group gatherings. Therefore, those who participated in the study were originally at a high risk of falling.

From the analysis, having a history of falls was positively associated with participants who had been going out at least once pre-pandemic and with those who had been going out five or more times. Previous research revealed that after experiencing a fall, people lose confidence in walking and fear falling, leading to a reluctance to move [14]. Moreover, community-dwelling older adults feared falling in the house and being unable to walk in the future after the first SOE, when going out were severely restricted due to the pandemic-imposed SOE [22]. Therefore, the participants who had fallen in the past might have become more reluctant to move due to anxiety about falling, and some even fell.

Older adults living alone will likely have more opportunities to perform household chores than those living with their families. A previous study found that more than half of the falls among community-dwelling older adults occur at home [33]. In this study, those who lived alone moved more frequently to perform household chores and may have been at a higher fall risk. Moreover, during the pandemic, a previous study reported that community-dwelling older adults living alone were more isolated than those living with their families [34]. Therefore, it is possible that these participants were more likely to fall because they spent more time at home than pre-pandemic.

This study has implications for fall prevention for community-dwelling older adults. As the COVID-19 pandemic continues, we need to pay more attention toward maintaining opportunities for going out, especially in the case of community-dwelling older adults who were going out frequently pre-pandemic. In other words, a person who frequently goes out and has good muscle strength, but suddenly lowers their frequency of going out, is at a high risk of falling. Previous frailty prevention policies have recommended constructing indoor places inside facilities for community-dwelling older adults to interact with each other [35]. Therefore, we need to provide additional support, such as by setting up outdoor places to prevent the spread of COVID-19, for example, this can be achieved by creating a walking map with beautiful views and organizing a frail prevention class in the park. Furthermore, there are limited opportunities for community-dwelling older adults to be objectively assessed for physical function. In the future, in addition to promoting health checkups, we may need to give opportunities to provide frailty checks and fall risk assessments at outpatient clinics and city halls.

This study has the following strengths. First, we were able to distribute the questionnaires when it was generally challenging to conduct surveys due to the ongoing pandemic. Second, we conducted the survey during the COVID-19 pandemic so that we could learn about the situation participants were facing at that time, with little recall bias. Third, since we used the data collected pre-pandemic, we could examine the association between the changes in the frequency of going out before and during the pandemic and falls.

However, there are several limitations to this study. First, we categorized one or more falls as falls. Prior studies have categorized falls as fractures, medical care, or reluctant falls. However, the incidence of these falls is low, and there are limitations to conducting a quantitative study in a single region. Future studies should thus consider a larger sample size and include classifying falls as injuries or multiple falls. Second, because we used the same items as in the survey conducted in 2020, the items asking about the frequency of going out were grouped into 2–4 times. Therefore, it was impossible to examine changes in the frequency of going out in detail. Further research is thus needed to consider the association between specific changes in the frequency of going out and falls. Third, we could not measure objective indicators, such as actual physical activity, in this study. We believe that, in some cases, the participants could maintain or improve their physical activity levels by performing exercises inside their homes. Unfortunately, the pandemic made it challenging to conduct a survey using actual measurements. A survey using actual measurements and a questionnaire is therefore necessary in the future. Fourth, the survey was conducted in one town. The results of this study are, thus, likely to differ from those of urban areas located on flat lands with well-developed public transportation systems and easy accessibility. Therefore, the findings should be carefully interpreted, and more studies are needed in regions with other characteristics. Finally, this study could not elucidate the mechanism through which prevention of going out was associated with falls. We need to conduct further qualitative research to clarify how the lives of the community-dwelling older adults changed after they were prevented from going out, which led to their falls.

## 5. Conclusions

This study emphasizes the importance of maintaining opportunities to go out to prevent falls among community-dwelling older adults. We need to pay more attention to community-dwelling older adults who were frequently going out pre-pandemic. The findings suggest that they should not be overconfident in their muscle strength and should check their risk for frailty and falls in order to prevent falls.

## Figures and Tables

**Table 1 ijerph-20-02650-t001:** Participants’ characteristics and their association with incidence of falls: Univariate analysis (n = 381).

	Total			Falls						
	(n = 381)			Yes (n = 91)			No (n = 290)			*p*
	n	(%)	Mean ± SD [Range]	n	(%)	Mean ± SD [Range]	n	(%)	Mean ± SD [Range]	
[Date from 2020 survey]										
Age			74.4 ± 6.2 [65–93]			76.2 ± 6.6 [65–91]			74.0 ± 5.9 [65–93]	0.005 ^a^
Sex: Female	196	(51.4)		51	(56.0)		145	(50.0)		0.314 ^b^
Living arrangement										
Living alone	50	(13.5)		19	(21.6)		31	(11.0)		<0.001 ^b^
Living with spouse	185	(49.9)		26	(29.5)		159	(56.2)		
Living with other than spouse	136	(36.7)		43	(48.9)		93	(32.9)		
Need of daily life support										
No	342	(91.9)		73	(83.9)		269	(94.4)		0.006 ^b^
Yes, but not receiving it	19	(5.1)		9	(10.4)		10	(3.5)		
Yes	11	(3.0)		5	(5.7)		6	(2.1)		
Dissatisfaction with financial status: Yes	85	(22.8)		26	(29.9)		59	(20.6)		0.072 ^b^
Having a history of falls	92	(24.6)		51	(58.0)		41	(14.3)		<0.001 ^b^
Having a fear of falls	161	(42.5)		54	(59.3)		107	(37.2)		<0.001 ^b^
Decrease in frequency of going out compared to last year	59	(15.5)		21	(23.1)		38	(13.1)		0.022 ^b^
Subjective sense of good health	324	(86.9)		67	(75.3)		257	(90.5)		<0.001 ^b^
Capable of climbing stars	251	(67.3)		46	(52.3)		205	(71.9)		<0.001 ^b^
Capable of standing up from a chair	312	(83.4)		67	(76.1)		245	(85.7)		0.036 ^b^
Cognitive function impairment	154	(41.8)		40	(46.5)		114	(40.4)		0.317 ^b^
Oral function impairment	115	(31.0)		37	(43.0)		78	(27.4)		0.006 ^b^
Capable of going out by themselves	312	(83.9)		65	(74.7)		247	(86.7)		0.008 ^b^
Feeling disinterested in the past month: Yes	72	(19.4)		24	(26.7)		48	(17.0)		0.044 ^b^
Participation in community activities ^d^										
Volunteers’ group	51	(17.6)		11	(17.7)		40	(17.5)		0.971 ^b^
Group exercises	100	(33.1)		19	(30.6)		81	(33.8)		0.643 ^b^
Recreation group	110	(35.6)		26	(39.4)		84	(34.6)		0.468 ^b^
Group gathering for flail prevention	72	(24.0)		23	(35.9)		49	(20.8)		0.012 ^b^
[Data from 2021 survey]										
Using stairs or hills to leave home	110	(29.6)		35	(39.3)		75	(26.5)		0.021 ^b^
Medication ^d^										
Antihypertensive drugs	173	(51.0)		42	(53.8)		131	(50.2)		0.571 ^b^
Sleeping pills	40	(11.8)		12	(15.4)		28	(10.7)		0.263 ^b^
Diuretic	29	(8.6)		12	(15.4)		17	(6.5)		0.014 ^b^
Antidepressant or anti-anxiety drugs	7	(2.1)		1	(1.3)		6	(2.3)		1 ^c^
Subjective symptoms ^d^										
Difficulty in seeing	58	(15.9)		13	(14.9)		45	(16.2)		0.782 ^b^
Wobbling	19	(5.2)		7	(8.0)		12	(4.3)		0.172 ^c^
Knee or back pain	113	(31.0)		37	(42.5)		76	(27.3)		0.007 ^b^
Difficulty in hearing	62	(17.0)		20	(23.0)		42	(15.1)		0.088 ^b^
Headache	8	(2.2)		2	(2.3)		6	(2.2)		1 ^c^
[Data from 2020 and 2021 survey]										
Preventing from going out	86	(22.6)		25	(27.5)		61	(21.0)		0.2 ^b^

Note: Missing data were excluded from the analysis. Abbreviation: SD = standard deviations. ^a^ Welch’s *t*-test. ^b^ χ^2^ test. ^c^ Fisher’s exact test. ^d^ Multiple choices were allowed.

**Table 2 ijerph-20-02650-t002:** The frequency of weekly going out pre-pandemic and at the time of the first SOE (n = 381).

		Total (n = 381)		At the Time of the First SOE							
				Less than Once (n = 86; 22.6%)		Once (n = 133; 34.9%)		2–4 Times (n = 130; 34.1%)		≥5 Times (n = 32; 8.4%)	
		n	(%)	n	(%)	n	(%)	n	(%)	n	(%)
Pre-pandemic	Once	43	(100.0)	23	(53.5)	17	(39.5)	2	(4.7)	1	(2.3)
	2–4 times	184	(100.0)	38	(20.6)	78	(42.4)	64	(34.8)	4	(2.2)
	≥5 times	154	(100.0)	25	(16.2)	38	(24.7)	64	(41.6)	27	(17.5)

Note: Percentage are calculated based on the frequency of weekly going out pre-pandemic. Abbreviation: SOE: state of emergency.

**Table 3 ijerph-20-02650-t003:** Incidence of falls; by frequency of weekly going out (n = 91).

		Total (n = 91)		At the Time of the First SOE							
				Less than Once (n = 25; 27.4%)		Once (n = 29; 31.9%)		2–4 Times (n = 32; 35.2%)		≥5 Times (n = 5; 5.5%)	
		n	(%)	n	(%)	n	(%)	n	(%)	n	(%)
Pre-pandemic	Once	13	(100.0)	8	(61.5)	5	(38.5)	0	(0.0)	0	(0.0)
	2–4 times	43	(100.0)	8	(18.6)	15	(34.9)	20	(46.5)	0	(0.0)
	≥5 times	35	(100.0)	9	(25.7)	9	(25.7)	12	(34.3)	5	(14.3)

Note: Percentage are calculated from the frequency of the pre-pandemic weekly going out. Abbreviation: SOE: state of emergency.

**Table 4 ijerph-20-02650-t004:** Factors of incident of falls; multivariate logistic regression analysis (n = 237).

	Incidence of Falls. (1: Yes 0: No)		
	AOR	95%CI	*p*
[Data from 2020 and 2021 survey]			
Preventing from going out (ref. Continuing going out)	1.95	[0.76–5.04]	0.167
[Data from 2020 survey]			
Living arrangement (ref. Living alone)			
Living with spouse	0.17	[0.05–0.61]	0.006
Living with other than spouse	0.24	[0.07–0.86]	0.028
Having a history of fall (ref. No)	14.82	[5.78–38.00]	<0.001

Note: Abbreviations: ref: reference, AOR: adjusted odds ratio, 95%CI: confidence interval. This model used force methods and included age, sex, and variables that were significant at *p* < 0.05 in univariate analysis (need of daily support, having fear of falls, subjective sense of health, decrease in frequency of going out compared to last year, capable of climbing stairs, capable standing up from a chair, oral function impairment, capable of going out by themself, feeling disinterested, participating group gathering for frail prevention, using stairs and hill to leave home, taking diuretic medication, and feeling knee or back pain).

**Table 5 ijerph-20-02650-t005:** Factors of incident of falls among those weekly going out five or more times; multivariate logistic regression analysis (n = 120).

	Incidence of Falls (1: Yes, 0: No)		
	AOR	95%CI	*p*
[Data from 2020 and 2021 survey]			
Preventing from going out (ref. Continuing going out)	6.84	[1.51–31.02]	0.013
[Data from 2020 survey]			
Living arrangement (ref. Living alone)			
Living with spouse	0.08	[0.02–0.40]	0.002
Living with other than spouse	0.15	[0.03–0.73]	0.020
Having history of fall (ref. No)	7.35	[1.81–29.84]	0.005
Participating in group gathering for frail prevention (ref. No)	6.09	[1.48–25.12]	0.013

Note. Abbreviations: ref: reference, AOR: Adjusted Odds Ratio, 95%CI: confidence interval. This model used force methods and included age, sex, and significant variables at *p* < 0.05 in univariate analysis (capable of climbing stairs and fear of falling).

## Data Availability

The datasets generated and analyzed during the current study are not available due to privacy or ethical restrictions.

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
