# Peer review of "Association between Prevention from Going Out and Incidence of Falls among Community-Dwelling Older Adults during COVID-19 Pandemic"

_ijerph, 2023, doi:10.3390/ijerph20032650_

Round 1
Reviewer 1 Report
Dear authors,
congratulations for your work. You tried to understand possible associations of falls with pandemic Covid-19 isolation. In my oppinion the article needs several improvemments to be considered to publication in IJERPH.
Major: Format the abstract and methods section according to IJERPH; and improve the quality of your tables that are not so clear.
1.ABstract: Add data in the methods;
2. Methods: Add study design; Add inclusion and exclusio criteria; add references to support your data collection;
3. Results: Improve your tables
regards
Reviewer 2 Report
The research makes an contribution to the literature. After taking into account some significant fixes.
- The abstract is well organized and contains the most relevant information.Introduction - correct
- I suggest that the methodology section be improved and that the authors break it down into more specific subchapters. There is a lack of description of some important parts, namely the inclusion / exclusion criteria from the research.
- Data analysis and results are correct.
- The discussion is well written. I suggest that the authors further develop the implications of this study for practice. I propose to add strengths and weaknesses.
- Strengths and limitations
- The list of bibliographic references is current and correct.
7. The criterion for inclusion in the study was: age over 60?/65, …. The criterion for exclusion from the study was: age below 60?/65,
8. Conclusions - were written very generally in relation to the obtained results. I propose to expand, to present in detail.
Dear Author/s,
I appreciate the enormous amount of work you have contributed to the submitted article.
Round 2
Reviewer 1 Report
Dear authors,
congratulations for your revision. You have attended my suggestions although there are minor revisions needed:
1:Abstract-Remove the number before backgroud, methods, results,conclusions....
2:Add the code number of ethics commitee
Author Response
Dear Reviewer
We would like to express our gratitude for the insightful comments, which have helped us improve our paper. We have revised our manuscript in accordance with these comments, as explained below.
1:Abstract-Remove the number before backgroud, methods, results,conclusions....
Response: We removed the number and headings in the abstract.
2:Add the code number of ethics committee
Response: We added the approval code. We have also added to the statement what we asked in the questionnaire about survey participation.
We also revised a few typographical errors.